# Nanofiber Graft Therapy to Prevent Shoulder Stiffness and Adhesions after Rotator Cuff Tendon Repair: A Comprehensive Review

**DOI:** 10.3390/biomedicines12071613

**Published:** 2024-07-19

**Authors:** Jong Pil Yoon, Hyunjin Kim, Sung-Jin Park, Dong-Hyun Kim, Jun-Young Kim, Du Han Kim, Seok Won Chung

**Affiliations:** 1Department of Orthopedic Surgery, School of Medicine, Kyungpook National University, Daegu 41944, Republic of Korea; altjp1@gmail.com (J.P.Y.); miniself@hanmail.net (S.-J.P.); kdh8110@hanmail.net (D.-H.K.); 2Department of Orthopedic Surgery, School of Medicine, Catholic University, Daegu 38430, Republic of Korea; dr.junyoung@gmail.com; 3Department of Orthopedic Surgery, Keimyung University Dongsan Hospital, Keimyung University School of Medicine, Daegu 42601, Republic of Korea; osmdkdh@gmail.com; 4Department of Orthopedic Surgery, Konkuk University Medical Center, Seoul 05030, Republic of Korea; smilecsw@gmail.com

**Keywords:** rotator cuff tears, shoulder stiffness and adhesions, nanofiber scaffolds, tissue engineering, clinical trials

## Abstract

Stiffness and adhesions following rotator cuff tears (RCTs) are common complications that negatively affect surgical outcomes and impede healing, thereby increasing the risk of morbidity and failure of surgical interventions. Tissue engineering, particularly through the use of nanofiber scaffolds, has emerged as a promising regenerative medicine strategy to address these complications. This review critically assesses the efficacy and limitations of nanofiber-based methods in promoting rotator cuff (RC) regeneration and managing postrepair stiffness and adhesions. It also discusses the need for a multidisciplinary approach to advance this field and highlights important considerations for future clinical trials.

## 1. Introduction

Rotator cuff tears (RCTs) are a major cause of shoulder dysfunction and pain, affecting approximately 21% of individuals. Arthroscopic rotator cuff repair (ARCR), commonly targeting the supraspinatus tendon, is a widespread procedure [1,2,3]. While it yields satisfactory functional outcomes in about 90% of cases [4,5], complications such as rotator cuff (RC) recession and postoperative stiffness are notable. The variability in the incidence of RC re-tears postsurgery, ranging from 20% to 90%, suggests a significant rate of healing failure, which is often associated with postoperative adhesions and capsular contracture [6,7]. Additionally, shoulder stiffness is a common issue following ARCR, with incidence rates between 2.3% and 28.3% [8,9,10]. Patients experiencing this stiffness face difficulties in pain management and in regaining shoulder range of motion (ROM). This condition, primarily resulting from contracture of the shoulder capsule and adhesion formation, is linked to a dysregulated fibrotic inflammatory response [11,12,13]. Notably, shoulder stiffness usually develops within the first three months postsurgery, affecting 11% to 35.4% of patients [9,14]. Despite successful surgical outcomes, this complication can lead to patient distress and dissatisfaction [15,16,17].

Shoulder stiffness is also frequently observed following RC surgical repair and notably impairs surgical success and recovery [18]. This complication contributes significantly to morbidity and can result in the failure of surgical interventions if not addressed. The underlying causes of postoperative stiffness are complex and multifactorial, including bursal inflammation, muscle contracture, and atrophy, alongside pain and weakness from damaged RC tendons, particularly in cases of adhesive capsulitis [19,20]. In patients with adhesive capsulitis, the likelihood of developing postoperative stiffness is estimated at 15% [18]. Furthermore, fibrosis and adhesions in the torn RC exacerbate pain, complicating postoperative rehabilitation and increasing the risk of persistent stiffness, thereby challenging the success of surgical management [21,22]. Various techniques have been rigorously examined for their effectiveness in repairing torn RC tendons and addressing the subsequent stiffness and adhesions. These include a range of suturing methods, surgical interventions [23,24], and tissue transplantation using allografts, xenografts, and autografts. Additionally, the use of decellularized constructs and tissue regeneration strategies has been explored [25,26]. Despite their widespread clinical use, current surgical approaches such as suturing and tissue grafting show limited efficacy in managing large RCTs and their associated complications. This limited success highlights the need for regenerative tissue strategies to improve RC healing. Such approaches involve cell therapy, growth factors, and biomaterial matrices, used either individually or in combination. Of particular interest are biomimetic matrices that mimic the mechanical and physiological properties of RC tendons, showing substantial potential in this field [27,28]. A schematic illustration of the normal structure of the rotator cuff and the areas where problems occur is shown in Figure 1.

The literature review begins with an examination of the current strategies employed to manage stiffness and adhesions following RCTs. It then details the roles and mechanisms of nanofibers in modulating cellular responses, controlling inflammatory and fibrotic reactions, remodeling the extracellular matrix (ECM), and providing physical barriers. The review also covers design considerations for nanofiber-based methods, drawing on insights from clinical trials and commercial applications. The primary focus of this discussion was to evaluate the effectiveness and potential of nanofiber-based structures as matrices that promote RC healing while effectively managing postrepair stiffness and adhesions. Moreover, the challenges and considerations associated with clinical trials and regulatory processes are discussed.

## 2. Mechanisms and Challenges in Managing Stiffness and Adhesions after Rotator Cuff Repair

### 2.1. Mechanism of Stiffness and Adhesion Formation

Throughout the healing process of RCTs, several mechanisms contribute to the development of stiffness and adhesions, underscoring the importance of understanding these processes to devise effective management strategies. Initially, an inflammatory response occurs after injury, characterized by the release of cytokines and growth factors that stimulate fibroblast proliferation and collagen deposition. However, prolonged or excessive inflammation can lead to scar tissue formation and adhesions, thereby exacerbating stiffness. Additionally, fibrosis plays a crucial role in tissue healing, with excessive collagen deposition impeding the smooth sliding of the tendon within its sheath, thus promoting ankylosis. Furthermore, tissue remodeling, essential for restoring structural integrity, can become dysregulated, leading to the formation of scar tissue and adhesions, which further impede shoulder joint mobility and increase stiffness [29,30]. Notably, adhesions and stiffness reciprocally influence each other’s symptoms; abnormal tissue attachments disrupt normal tendon glide, worsening stiffness [31]. Adhesions can be classified as congenital or acquired (postinflammatory or postoperative) depending on their etiology. Postoperative adhesion formation involves the inhibition of fibrinolysis and ECM decomposition systems, triggering an inflammatory response associated with the production of cytokines and transforming growth factor-beta (TGF-β1), and inducing tissue hypoxia through disrupted blood delivery to fibroblasts, leading to the expression of hypoxia-inducible factor-1α and vascular endothelial growth factor (VEGF) [32,33]. Understanding these complex and interconnected mechanisms underlying stiffness and adhesion formation is critical for implementing a comprehensive management approach, which includes surgical interventions, rehabilitation protocols, and targeted strategies to address inflammation, promote tissue remodeling, and prevent adhesion formation [34].

### 2.2. Current Treatment Techniques for Stiffness and Adhesions

Postoperative treatment options following RCTs repair involve a variety of interventions tailored to meet individual patient needs. The primary approach for severe injuries is surgical intervention, which includes reattaching the torn tendon to the bone, facilitating tendon healing, and restoring shoulder function using techniques such as arthroscopic repair or open surgery [34]. Complementary nonsurgical modalities enhance a structured rehabilitation regimen focused on strengthening and flexibility. This includes the use of pharmacological agents for pain and inflammation management, such as nonsteroidal anti-inflammatory drugs (NSAIDs), and adjunctive methods like extracorporeal shock wave therapy to optimize recovery and prevent re-injury [35,36]. Additional options include biological regeneration or augmentation strategies that utilize stem cell therapy and allogeneic or autologous tendon transplantation techniques [37,38].

However, the management of stiffness and adhesions after RCTs faces several notable limitations within current practices. These methods often do not replicate the natural tendon healing process and neglect the crucial fusion zone where the four RC tendons converge at the humeral insertion. Furthermore, they may lack the necessary mechanical strength or face potential biological rejection postrepair. Current techniques predominantly rely on sutures and bone anchors to connect tendons and bones. Although frequently addressing adhesions and ankylosis, these methods can impede successful healing and rehabilitation, thereby prolonging recovery times, reducing treatment effectiveness, and increasing the risk of re-injury. Complications may include persistent pain, tendon compromise, and postoperative dysfunction. Recognizing these challenges, there is growing interest in employing nanofiber-based strategies to enhance the stiffness and adhesion properties of RC healing. By addressing these issues, surgeons can improve the healing trajectory, restore shoulder function, and ultimately enhance patient outcomes after RC reconstruction [39].

## 3. Characteristics and Mechanisms of Nanofiber-Based Approaches in Tissue Engineering

### 3.1. Characteristics of Nanofiber Scaffolds

Nanofiber-based approaches play several critical roles in tissue engineering. These include providing a biomimetic structure for cell attachment [40], facilitating drug and protein delivery [41], and enhancing the mechanical properties of the scaffolds [42]. Additionally, they regulate cellular responses [43], manage inflammation [44], and guide tissue organization and function during healing processes [45].

#### 3.1.1. Structural and Morphological Properties

Polymeric nanofibers are essential in tissue engineering due to their unique properties. The term “nanofiber” refers to fibers with diameters between 1 and 1000 nanometers [46]. Their small diameter, similar to that of ECM fibers, makes them excellent biomimetic scaffolds [47,48,49]. Additionally, their high surface area-to-volume ratio enhances cell attachment [50] and facilitates drug loading [51]. Notably, nanofibers are critical for cell adhesion to biomaterial surfaces due to their rapid protein adsorption rates. For instance, poly(l-lactic acid) (PLLA) surfaces with diameters between 50 and 500 nanometers have been shown to adsorb proteins four times more effectively than porous PLLA structures [52]. Polymer nanofibers also exhibit distinct mechanical properties. Research has shown that tensile modulus [53,54], tensile strength [55], and shear modulus [56] increase as the fiber diameter decreases. This is attributed to enhanced alignment of macromolecular chains within the fiber as the diameter reduces [57]. Smaller diameter nanofibers display increased crystallinity, likely due to flow-induced crystallization during the electrospinning process. These mechanical properties not only influence cell behavior but also provide the necessary tension and strength to resist cytoskeletal forces [52].

#### 3.1.2. Compositional Properties

The chemical composition of nanofibers is crucial for their suitability in tissue engineering. Polymers like PLLA, PCL, and collagen are commonly used due to their biodegradability and biocompatibility. PLLA degrades into lactic acid, minimizing toxicity [58], while PCL offers a slower degradation rate, providing prolonged support [59]. Collagen supports cell adhesion and proliferation [60]. The mechanical properties, such as tensile strength and elasticity, are influenced by the polymer choice; PLLA offers high tensile strength, PCL provides flexibility, and collagen mimics the extracellular matrix (ECM). The degradation rate of nanofibers must match the application requirements to avoid residual inflammation [61]. The polymer’s chemical properties affect scaffold-cell interactions; collagen promotes cell attachment, while synthetic polymers like PLLA and PCL can be functionalized to enhance bioactivity. Surface modifications, such as plasma treatment, improve the bioactivity of synthetic nanofibers [62]. Polymer blends and composites, like PLLA-PCL mixtures, balance strength and flexibility [63], and adding bioactive materials like hydroxyapatite enhances tissue regeneration [64]. In summary, careful selection and combination of polymers create tailored nanofiber scaffolds that enhance tissue regeneration, ensure biocompatibility, and provide mechanical support, broadening regenerative medicine possibilities.

#### 3.1.3. Physico-Chemical Behavior

The physico-chemical behavior of nanofibers, including their hydrophilicity, porosity, and surface charge, significantly impacts their performance in tissue engineering. Hydrophilicity can be tailored to enhance cell attachment and proliferation by modifying the surface with bioactive molecules or coatings [65]. For example, introducing hydrophilic groups or applying coatings of proteins like fibronectin and laminin can improve cell adhesion and growth [66]. The porosity of nanofiber scaffolds influences nutrient diffusion and waste removal, which are crucial for maintaining cell viability within the scaffold. High porosity ensures effective mass transport, providing cells with sufficient nutrients and oxygen while allowing for the removal of metabolic waste [67]. Additionally, the pore size and distribution within the scaffold play a role in cell infiltration and tissue integration, with optimal pore sizes varying depending on the specific tissue engineering application [68]. Surface charge affects protein adsorption and cell behavior, with positively charged surfaces generally enhancing cell adhesion due to electrostatic interactions with negatively charged cell membranes [69]. By manipulating these physico-chemical properties, nanofiber scaffolds can be engineered to create a favorable microenvironment for tissue regeneration, promoting cell viability, proliferation, and differentiation. The key physicochemical parameters for nanofiber scaffolds are shown in Table 1.

#### 3.1.4. Fabrication Techniques

Various methods for fabricating nanofibers have been explored, including phase separation, self-assembly, 3D printing, and electrospinning [52]. Electrospinning, in particular, has gained significant attention for its simplicity, efficiency, and cost-effectiveness [78]. It has expanded rapidly, benefiting from its ability to handle a wide range of materials such as ceramics, polymers, and composites [79]. The technology encompasses several techniques like solution electrospinning and other variants such as emulsification electrospinning, mixed electrospinning, and coaxial/triaxial electrospinning, depending on the material state and desired fiber properties [80]. This versatility allows for the integration of inorganic fibers, bioactive agents, and polymer solutions, enabling the production of core-shell structures or hollow fibers, or the functionalization of fibers through specific coatings [81,82].

### 3.2. Mechanism of Nanofiber Scaffolds for Rotator Cuff Healing

#### 3.2.1. Regulation of Cell Response

Nanofibrous scaffolds are integral to regulating responses during RC healing, particularly in directing tissue organization and function. The mechanisms involved are illustrated in Figure 2. Tendons and ligaments experience unidirectional mechanical loads, leading to highly anisotropic mechanical properties due to extensive ECM fiber alignment. This anisotropy, which features tensile properties in the direction of fiber alignment significantly surpassing those in the perpendicular direction, enhances cell attachment—especially of mesenchymal stem cells and fibroblasts—and subsequently dictates cell alignment and behavior. Aligned nanofiber structures have shown promise as scaffolds for tendon regeneration, facilitating the spatial organization of cells and ECM deposition, thus supporting tissue maturation and functionality [52].

#### 3.2.2. Controlled Release of Bioactive Factors and Regulation of Local Inflammatory Response

Additionally, nanofibrous scaffolds are critical in managing inflammatory and fibrotic responses postRCTs. By enabling the controlled release of bioactive molecules and growth factors, these scaffolds modulate the local inflammatory environment, fostering an immune response that supports effective tissue healing. Anti-inflammatory cytokines mitigate excessive inflammation, whereas growth factors such as TGF-β promote fibroblast activation and collagen synthesis, which are vital for tissue repair. Moreover, nanofibrous scaffolds serve as reservoirs for the release of therapeutic agents, further influencing the inflammatory and fibrotic pathways to enhance healing outcomes [83].

#### 3.2.3. ECM Remodeling

Furthermore, nanofibrous scaffolds support ECM remodeling, which is essential for restoring tissue mechanics and function following RC healing. Acting as structural templates for ECM deposition and organization, nanofibers direct collagen fiber alignment, aiding in tissue maturation and improving biomechanical properties. This dynamic remodeling process, marked by the realignment of collagen fibers in response to mechanical loads, plays a crucial role in reinstating tissue strength, elasticity, and functionality, thus equipping the healed RC to endure physiological stresses [84,85].

#### 3.2.4. Adhesion and Stiffness Management

Nanofibers present a promising approach to addressing stiffness and adhesion following RCTs, primarily due to their ability to form barriers and perform hemostatically. By serving as physical barriers, nanofibers effectively prevent the formation of adhesions during the healing process, critically inhibiting the development of fibrous adhesions that are often linked with stiffness and restricted mobility postinjury. The engineered surface properties of these nanofibers enhance their barrier function by promoting cell repulsion and maintaining low surface energy, which further reduces the likelihood of tissue adhesion. Additionally, the hemostatic properties of nanofibers play a crucial role in controlling bleeding from damaged vessels, promoting hemostasis, and preventing hematoma formation—a known precursor to stiffness and adhesion. Nanofibers also absorb excess fluid and blood, thereby creating an optimal healing environment that reduces the accumulation of inflammatory exudates and fibrin, factors known to contribute to the formation of adhesions. Furthermore, the biocompatibility and biodegradability of nanofibers ensure their safe integration into the body over time, minimizing the risk of long-term complications, including stiffness. In summary, nanofibrous scaffolds offer a multifaceted approach to mitigate adhesions and stiffness after RCTs. They enhance cellular responses, modulate inflammatory and fibrotic reactions, facilitate ECM remodeling, and provide both physical barrier and hemostatic functions, establishing their significance in tissue engineering and regenerative medicine [85,86].

### 3.3. Improvement of Rotator Cuff Adhesion through Preclinical Cases

Several studies have underscored the therapeutic potential of nanofiber scaffolds in minimizing adhesions following RCTs. For instance, Romeo et al. [87] demonstrated Sharpey fiber-like attachment—a marker of improved mechanical and histological quality at the tendon-bone interface—in a sheep model of acute RCTs using electrospun polyglycolic acid (PGA) and poly(L-lactide-co-3-caprolactone) (PLCL)-based nonwoven microporous nanofiber matrices. Similarly, Han et al. [88] explored the impact of topical recombinant human parathyroid hormone (rhPTH) on tendon-bone healing in a rabbit model, utilizing electrospun and 3D-printed polycaprolactone (PCL) and hyaluronic acid (HA)-based nanofiber sheets. Their findings indicated elevated levels of collagen type I alpha and enhanced maturity of the tendon-bone junction. Additionally, Chen et al. [89] examined the preventive capabilities of HA/ibuprofen (IBU) and 1,4-butanediol diglycidyl ether (BDDE)-based multifunctional nanofibrous membranes on fibroblast adhesion and infiltration in a rabbit flexor tendon rupture model. These membranes served as effective physical barriers to fibroblast invasion and showed efficacy in reducing local inflammation and preventing tendon adhesions. Wei et al. [90] also investigated the hemostatic efficiency of self-assembled RATEA16 peptide nanofibers in a rabbit bleeding model, attributing the rapid hemostasis to the nanofiber network structure.

Collectively, these studies highlight the effectiveness of nanofiber-based scaffolds in enhancing RC healing and reducing adhesion by modulating various cellular and tissue responses. However, it is important to note that animal models do not completely mirror the human injury condition. The rat shoulder model is commonly employed to evaluate the initial safety, mechanisms, and efficacy of biological therapies aimed at tendon-to-bone repair, while larger animal models are preferred for studying surgical interventions. Ultimately, assessing the safety and efficacy of mechanical or combined biological/mechanical strategies in human patients remains a critical step.

## 4. Nanofiber-Based Strategies for Managing Stiffness and Adhesions in Rotator Cuff Healing

### 4.1. Design Considerations for Nanofiber Scaffolds

#### 4.1.1. Overview of Design Considerations

Despite the biomimetic properties and ECM-like structures of electrospun nanofiber scaffolds, their use in RC recovery and the management of stiffness and adhesions requires careful consideration. Nanofibers must act as a physical barrier to minimize tendon adhesion to the surrounding sheath, reduce tendon adhesion risk by delivering anti-adhesion and anti-inflammatory agents locally, and maintain enough strength and adhesion to provide cohesion during RC recovery and mechanical support akin to that of the native tendon. These considerations are summarized in Figure 3 [39,91].

#### 4.1.2. Anti-Adhesion Properties

In managing postoperative tendon adhesions, it is critical that the barrier not only reduces fibroblast adhesion but also provides a lubricating effect to ensure smooth tendon glide without hindering ligament cell proliferation. Excessive fibroblast proliferation and protein synthesis can lead to dense connective tissue formation, which in turn causes adhesions that restrict joint movement, resulting in tendon slippage and increased pain. Materials known for their anti-surface adhesion properties, such as PLA, PCL, and HA, are beneficial [92,93]. Alternatively, developing a multilayer membrane with enhanced surface anti-adhesion capabilities could further improve the efficacy of the physical barrier [94,95].

#### 4.1.3. Piezoelectric Materials

Porous electrospun fibrous membranes serve as a versatile platform for the sustained release of therapeutic agents, including anti-inflammatory and anti-adhesive agents, and biological factors such as genes and growth factors, into the target tendon. Implementing a core-shell structure, which segregates the membrane into outer and inner layers, facilitates biological factor release while acting as a physical barrier against fibroblast invasion [91,96]. Additionally, materials like assembled metal-organic frameworks (MOFs), known for their large pore volumes, high surface areas, and hydrophilicity, are promising for tendon regeneration [97]. Piezoelectric materials, such as graphene and Mxene, provide motion-driven mechanical stimulation [98], and their synergistic integration with materials like MoS2 can reduce inflammation and adhesion formation and exhibit antibacterial effects by suppressing reactive oxygen species (ROS) [99,100]. Exploring these material combinations and integration strategies is essential for enhancing tendon regeneration and reducing postoperative complications.

#### 4.1.4. Based on Polymer Types

Nanofibers can be engineered to perform specific functions in tendon repair. Some nanofibers are designed to form a physical barrier that reduces fibroblast adhesion, thereby preventing the formation of adhesions and scar tissue that can limit tendon mobility [91]. This anti-adhesive property is particularly useful in maintaining the functional integrity of the tendon during the healing process. On the other hand, natural polymer-based nanofibers can enhance the adhesion and proliferation of tenocytes and stem cells [101]. These fibers serve as a scaffold that supports cellular activities essential for tissue regeneration, promoting effective tendon repair. By tailoring the surface properties and composition of nanofibers, it is possible to achieve the desired balance between preventing unwanted adhesions and facilitating tissue regeneration.

#### 4.1.5. Enhancing Mechanical Properties

Natural polymer-based fibers, which resemble native tendons, enhance cell adhesion, proliferation, and differentiation. Despite their benefits, these fibers exhibit low mechanical strength, which can be a significant limitation. To address this issue, mechanically robust materials such as silk and insoluble collagen are often used either alone or in combination with synthetic polymers. These enhancements improve both mechanical strength and tissue regeneration [102,103,104]. Moreover, to mitigate mechanical weaknesses, integrating nanofibers with diverse structures—such as fabric or multiscale structures through nanofiber strands and 3D printing techniques—can emulate natural collagen fibers and bolster mechanical properties [105,106,107].

#### 4.1.6. Alternative Fixation Methods

A possible reason for the high failure rate in tendon-bone repair surgeries might be that the initial repair strength is insufficient to prevent gapping or rupture. Traditional sutures tend to concentrate shear stresses at a few anchor points, which does not adequately disperse the load across a broader attachment area. As a promising alternative, adhesive film-based methods are attracting increased interest [108]. An illustrative example is the bilayer Janus patch, characterized by anisotropic tissue adhesiveness. This patch features an inner adhesive layer that serves as a factor delivery platform and an outer layer that provides mechanical support while preventing the invasion of surrounding cells or tissues [108,109]. Additionally, enhancing the adhesive interactions of polymers enables easy application to moist tissues without sacrificing biocompatibility. These polymers can also be endowed with thermoresponsive and injectable properties, which increase their field applicability and potential to replace invasive mechanical fixation [110,111]. Ultimately, the effective design of nanofibers to prevent adhesions and stiffness following RCTs is crucial for promoting successful recovery and minimizing postoperative complications.

### 4.2. Clinical Trials for Rotator Cuff Healing and Adhesion Management

Clinical studies and case series have explored the efficacy of various scaffold materials in enhancing rotator cuff (RC) healing and managing adhesions postsurgery. A prospective, multicenter clinical trial by Barbash et al. [112] evaluated the use of BioFiber, a bilayer absorbable scaffold, in arthroscopic RC repair. The study demonstrated that BioFiber augmentation improved repair integrity and functional outcomes, with a 96% repair success rate observed via ultrasound at six months postoperation. Another randomized trial by Ferreira De Barros [113] investigated the use of a bioinductive porcine collagen scaffold, showing significant improvements in functional scores and a reduced retear rate compared to standard repairs. Additionally, a study by Beleckas et al. [114] described the Rotium wick, an FDA-approved interpositional nanofiber scaffold, which enhanced cellular organization and tendon strength, resulting in a 91% tendon healing rate and notable improvements in patient outcomes. A case series by Seetharam et al. [115] further supported the efficacy of the Rotium wick, reporting significant improvements in functional outcomes and a high rate of tendon healing in patients with small to medium RC tears. Another study by Beleckas et al. [116] examined short-term radiographic and clinical outcomes of RC repair augmented with an inter-positional nanofiber scaffold, showing improved outcomes compared to standard repair methods. Lastly, Cai et al. [117] conducted a randomized controlled study on the augmentation of arthroscopic RC repair using three-dimensional biologic collagen for moderate to large tears and demonstrated significant improvement in functional scores and a reduction in recurrence rate to 13.7%. These studies collectively indicate that scaffold augmentation can provide mechanical support and biological enhancement, contributing to better RC repair outcomes and adhesion management. Figure 4 outlines the flow chart of the literature search process, documenting the identification and selection of relevant studies. Table 2 shows examples of scaffold materials used in clinical settings following rotator cuff tears.

### 4.3. Commercialization Cases for Tendon Reconstruction and Adhesion Prevention

Table 3 highlights products that the FDA 510(k) has approved for tendon reconstruction and adhesion prevention, emphasizing their efficacy in enhancing soft tissue strength [118]. The scaffolds approved for commercial use primarily consist of type I collagen, HA, and PLA, which offer structures that facilitate cell proliferation and tissue growth while exhibiting bioactive, anti-adhesion, and bioabsorbable properties. Notable tendon augmentation scaffolds featured in clinical trials include TAPESTRY^®^ [119], GTR^®^ [120], and TenoMedTM^®^ [121]. Additionally, anti-adhesion films like Interceed^®^ [122], Seprafilm^®^ [123], and DK-film^®^ [124] have also received significant recognition. Scaffolds produced via electrospinning mimic the ECM, enhancing cell adhesion, tissue regeneration, and offering structural advantages and biocompatibility essential for tendon repair [124]. For instance, TAPESTRY^®^, developed by Embody, Inc., consists of collagen type I and poly (D, L-lactide) (PDLA), promoting ligament tissue regeneration and forming collagen connective tissues upon complete in vivo absorption [125]. Another notable product, J&J MedTech’s Interceed^®^, is an absorbable anti-adhesion membrane made from oxidized regenerated cellulose, noted for its bacteriostatic properties that effectively prevent adhesions by forming a protective barrier for 5–7 days without causing an inflammatory response [122].

The introduction of various products, primarily through clinical trials leading to commercialization, offers promising prospects in tendon repair and adhesion prevention. Products such as TAPESTRY^®^, GTR^®^, and TenoMed^®^ are designed to support cell growth and tissue regeneration. Complementing these are anti-adhesion films like Interceed^®^, Seprafilm^®^, and DK-film^®^, which are recognized for their effectiveness in preventing unwanted tissue adhesions and improving surgical outcomes. However, despite these advancements, challenges remain. Some commercially available products have been linked to complications, including inflammatory reactions, adhesion formations, and material rejections. These issues underscore the ongoing necessity for innovative solutions to enhance current methods and develop superior products for both tendon reconstruction and adhesion prevention.

## 5. Challenges and Future Directions

Tissue engineering strategies that employ nanofiber-based scaffolds show potential in enhancing tendon healing and reducing stiffness and adhesions following RC repair. Preclinical studies highlight the ability of nanofibers to mimic native tendon properties and effectively reduce adhesions, inflammation, and fibrosis, demonstrating the significant promise of this technology in managing postoperative complications. Nonetheless, our review of the current literature indicates an overreliance on simplistic models in the postoperative management of RCTs. These models often focus on isolated factors and fail to address the complex interactions involved in inflammatory responses, fibrosis, ECM remodeling, and tissue hypoxia. Importantly, the multifactorial nature of postoperative adhesions adds complexity to the development of consistent treatment or prevention strategies [33]. Challenges remain in scaffold design, including ensuring mechanical strength, managing inflammation and hemorrhage, optimizing tissue interactions, and maintaining organ function after transplantation. The clinical efficacy and safety of these scaffolds in humans remain uncertain, necessitating further research.

Efforts to apply nanofiber-based approaches to alleviate stiffness and adhesions in RCTs face significant regulatory challenges that impede their clinical adoption. The U.S. Food and Drug Administration (FDA) requires comprehensive preclinical testing, adherence to good manufacturing practice (GMP) standards, and thorough evaluations of safety, efficacy, and biocompatibility prior to the approval of clinical trials [126]. Given the broad range of applications and specific uses of tissue-engineered products, compliance with regulations is complex and depends on the components and configurations of these products. Critical to this process are the interactions between cells and scaffolds, which significantly affect the final product’s properties. It is crucial to understand the mechanisms of adhesion formation and their varying expression across different anatomical sites [127]. Moreover, balancing the physiological inflammatory responses within implanted scaffolds, assessing immune reactions, performing extensive preclinical studies using large animal models, and conducting detailed long-term in vivo safety evaluations are essential. Additionally, establishing objective, standardized criteria for evaluating adhesion severity will enable more meaningful comparisons and integration of research findings [32,33].

One of the key regulatory challenges involves the approval process for new tissue-engineered products. The FDA requires a multiphase testing protocol, starting with in vitro studies and progressing through animal models before human clinical trials. Each phase must demonstrate safety and efficacy, which can be a time-consuming and costly process. Furthermore, GMP compliance is essential to ensure product consistency and quality, but it adds another layer of complexity and expense to the development pipeline. There is also a need for standardization in evaluating the performance of these scaffolds. The variability in the types of polymers used, fabrication methods, and application techniques can lead to inconsistent results, making it difficult to draw definitive conclusions about their effectiveness. Establishing standardized testing protocols and evaluation criteria is crucial for advancing this field [128]. To successfully translate nanofiber-based strategies into clinical practice for RC healing, it is essential to address these regulatory hurdles. This involves not only meeting the stringent requirements set by regulatory bodies but also advancing our understanding of the underlying biological mechanisms and developing robust, reproducible methodologies. Collaborative efforts between researchers, clinicians, and regulatory agencies will be vital in overcoming these challenges and bringing effective new treatments to patients. Addressing these considerations is vital for the successful clinical translation of nanofiber-based strategies for RC healing [91].

## 6. Conclusions

In conclusion, nanofiber-based strategies present a promising method for addressing the challenges of stiffness and adhesion in RCTs. These strategies enhance therapeutic outcomes by modulating cellular responses, regulating inflammation and fibrosis, remodeling the ECM, and providing a physical barrier to achieve hemostasis. However, the complexity of these applications requires a multidisciplinary approach that integrates basic science, materials science, and clinical science. Successful scaffold design hinges on the integration of diverse materials and technologies, with careful consideration of the clinical setting to optimize tissue interactions and ensure long-term functional outcomes. Therefore, overcoming these multifaceted challenges, navigating regulatory frameworks, thoroughly reviewing preclinical results, and addressing in vivo safety concerns are critical steps toward the successful clinical application of nanofiber-based strategies for healing RCTs.

## Figures and Tables

**Figure 1 biomedicines-12-01613-f001:**
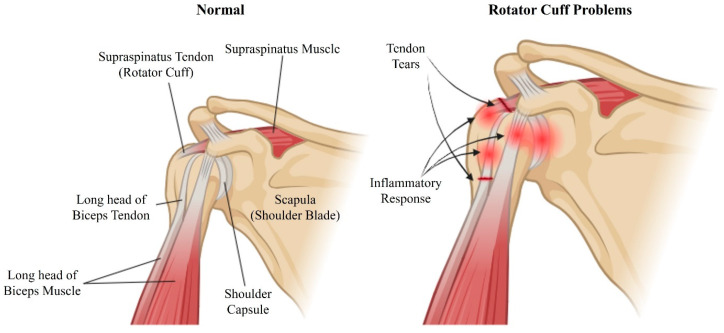
Schematic illustration of the shoulder anatomy highlighting normal structures (**left**) and regions affected by rotator cuff problems (**right**). The normal shoulder includes the supraspinatus tendon (rotator cuff), supraspinatus muscle, scapula (shoulder blade), long head of the biceps tendon, long head of the biceps muscle, and shoulder capsule. The problematic shoulder indicates tendon tears and the inflammatory response commonly associated with rotator cuff injuries. Created with BioRender.com.

**Figure 2 biomedicines-12-01613-f002:**
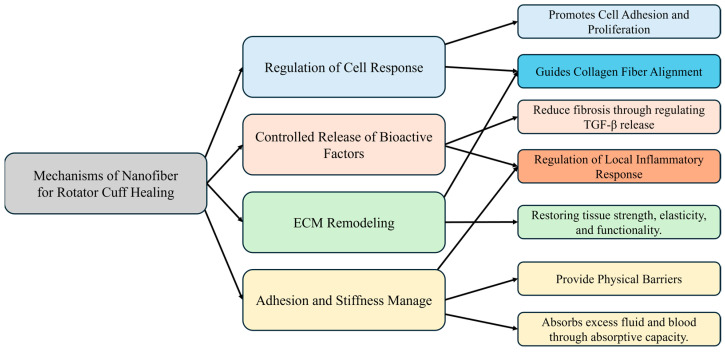
Tissue engineering mechanism of nanofiber scaffolds.

**Figure 3 biomedicines-12-01613-f003:**
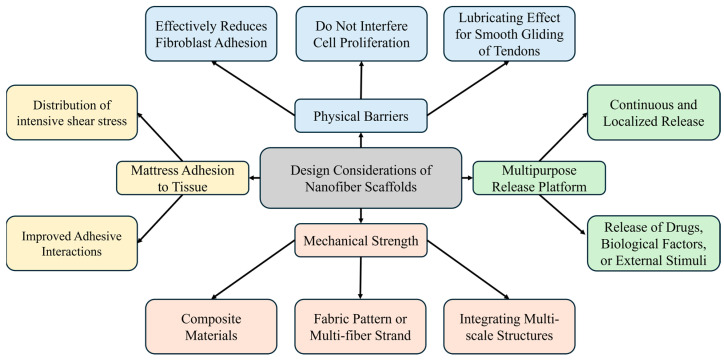
Design considerations for nanofiber scaffolds.

**Figure 4 biomedicines-12-01613-f004:**
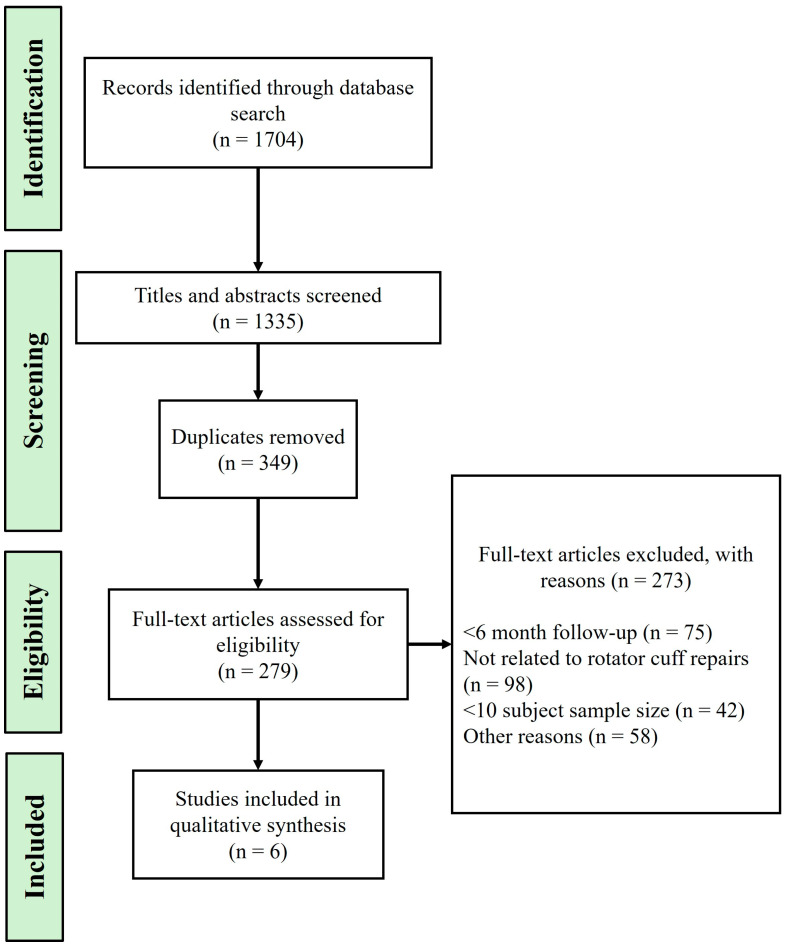
Flowchart of the literature search and selection process.

**Table 1 biomedicines-12-01613-t001:** Key Physicochemical parameters of Nanofiber Scaffolds.

Physicochemical Parameters	Optimal Range	Importance
Fiber Diameter	10–500 nm [70]	Influences surface area-to-volume ratio, cell attachment, and mechanical properties.
Porosity	80–90 % [67,68]	Affects nutrient diffusion, waste removal, and cell infiltration.
Pore Size	6–20 μm [68,71]	Essential for tissue integration and cell migration.
Surface Hydrophilicity	Water contact angle 35°–60° [65,72,73]	Enhances cell attachment and proliferation.
Mechanical Properties	Tensile Strength: 4.4–660 MPa [74,75] Elastic Modulus: 200–1500 MPa [74,75,76] Strain: ~35% [77]	Ensures scaffold integrity and mimics native tissue mechanics.
Biocompatibility	Nontoxic, nonimmunogenic [70]	Ensures safe integration with host tissue.

**Table 2 biomedicines-12-01613-t002:** Clinical cases involving the use of scaffold materials for tendon repair.

Author	Study	Scaffold Material	Study Type	Main Findings
Barbash et al. [112]	Clinical Outcomes and Structural Healing After Arthroscopic Rotator Cuff Repair Reinforced With A Novel Absorbable Biologic Scaffold	BioFiber, bi-layer absorbable scaffold	Prospective, multicenter clinical trial	Improved repair integrity and functional outcomes, 96% repair success rate at 6 months
Ferreira De Barros [113]	Bioinductive Scaffold Augmentation in Complete and Massive Rotator Cuff Tears	Bioinductive porcine collagen scaffold	Randomized trial	Significant improvements in functional scores, reduced retear rate
Beleckas et al. [114]	Rotator Cuff Repair Augmented With Interpositional Nanofiber Scaffold	Rotium wick, interpositional nanofiber scaffold	Study with technical note	Enhanced cellular organization and tendon strength, 91% tendon healing rate
Seetharam et al. [115]	Use of a Nanofiber Resorbable Scaffold During Rotator Cuff Repair	Rotium wick, interpositional nanofiber scaffold	Case series	Significant improvements in functional outcomes, high rate of tendon healing in small to medium RC tears
Beleckas et al. [116]	Short-Term Radiographic and Clinical Outcomes of Arthroscopic Rotator Cuff Repair with and without Augmentation with an Interpositional Nanofiber Scaffold	Interpositional nanofiber scaffold	Case series	Improved radiographic and clinical outcomes
Cai et al. [117]	Arthroscopic Rotator Cuff Repair With Graft Augmentation of Three-Dimensional Biological Collagen for Moderate to Large Tears	3D biological collagen	Randomized controlled study	Significant improvements in functional scores, reduced retear rate to 13.7%, better tendon-bone healing

**Table 3 biomedicines-12-01613-t003:** Commercially available products for tendon augmentation and adhesion prevention.

Product	Company	Compositions	Applications
TAPESTRY^®^ [119]	Embody, Inc. (Norfolk, VA, USA)	Collagen and PDLA	Tendon and ligament healing
GTR^®^ [120]	GTR BioTech. Co., Ltd. (Fuzhou, China)	Collagen separated from bovine tendon tissue.	Tendon healing
TenoMed^®^ [121]	Exactech, Inc. (Gainesville, FL, USA)	Absorbable type I collagen matrix	tendon healing and provide a sliding surface
Interceed^®^ [122]	Johnson & Johnson MedTech Co. (New Brunswick, NJ, USA)	Oxidized regenerated cellulose	Tendon and abdominal adhesion prevention and protective coating
Seprafilm^®^ [123]	Baxter International Inc. (Deerfield, IL, USA)	HA and carboxymethylcellulose (CMC) based	Abdominal and pelvic adhesion prevention
DK-film^®^ [124]	Chengdu Dickon Pharmaceutical Co. (Chengdu, China)	PLA based	Tendon and abdominal adhesion prevention

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
