# Peer review of "Nanofiber Graft Therapy to Prevent Shoulder Stiffness and Adhesions after Rotator Cuff Tendon Repair: A Comprehensive Review"

_biomedicines, 2024, doi:10.3390/biomedicines12071613_

Round 1

Reviewer 1 Report

Comments and Suggestions for Authors

The manuscript entitled “Nanofiber Graft Therapy ro Prevent Shoulders Stiffness and Adhesions after Rotator Cuff  Tendon Repair: A comprehensive Review” aims to present a comprehensive review of the NANITECMEC nanofiber graft therapy to prevent shoulder stiffness and adhesions after RCTR. The authors present s the main techniques discussing on surgical interventions, and some rehabilitation protocols. It was shown that some new nanofiber scaffolds could provide an efficient physician barrier to minimize the  tendon affection. In this sense, are mentioned electrospun nanofibres as members anes or scaffolds. The authors also included mentions on the complications of ARCR, RC and the stiffness.  They observations: 1. There was not discuss the lack of regulating compliance in the area of tissue engineering dedicated to the tendon healing, nor the domain of reducing stiffness and adherence after RCR. 2. The manuscript is not reader friendly, therefore I suggest the authors to restructure their manuscript using more headings and subheadings. Recommendations: a. The number 3.2 of subheading is repeated. Please correct. b. Under 4.1. should  be subheadings for: porous membranes; piezoelectric materials; natural polymer based nanofibers. c. Review section 2- in the title is mentioned “current strategies “ but, such a topic is not included. Please revise. d. Title of section 3 mentions “ Role and mechanism…” but, no specific, or clearly the role is presented. Please clarify. e. Sub section 3.1. - the title does not reflect the content. Through characterization it is generally understood the study of the characteristics as structure, morphology, composition, physico-chemical behavior, applying various investigation methods. Or, the authors presents it in 3.1. pages only a few aspects related to the fibers dimensions. Please reconsider the 3.1. Subsection and further develop it.  f. The title of 3.2. mention just “ Mechanism to nanofiber scaffolds” - to what mechanism do the authors refer? Please clarify. g. Present the clinical trials in a separate section and the commercialization cases in another section. h. Present the challenges and conclusions in separate sections. In such a way, it is possible to treat more accurately the challenges: materials, procedures, a.s.o. i. The world inclusion of specific examples or case studies would be beneficial for the review thus being more informative. 3. Although the manuscript transmits the need for more clinical trials, there are only few references to the already done clinical trials. Please consult the Cochrane database. 4. I strongly recommend the authors to do the PRISMA diagram. From such diagram it would be easier for everyone to understand the literature study performed, as well as the way the authors have interpreted and used it. 5. The caption for Table 1 should be changed to correctly reflect it’s contents. 6. Complete the references. Examples: a. Line 210- When mentioning Chen work, please include the corresponding reference. b. Line 291- same for Parker work c. Line 297- same for Pearson work etc. The present  manuscript needs further improvements and development prior its publication. 

Comments on the Quality of English Language

English language is fine. Minor editing issues to be addressed.

Reviewer 2 Report

Comments and Suggestions for Authors

The Authors review use of nanofibers in shoulder stiffness therapy. The owrk is well written and organized.However, I think it could benefit if Authors made some more amandements:

1) the schematic figure presentic the shoulder, pointing to the regions were problems occurs

2) the list of physicochemical parameters which are important, when designing the proper material for nanofibers, with the ranges of optimal values

3) In paragraph 4.1 the Authors state that nanofibers form barier reducing adhesion of fibroblasts, which is treated as their advantage. Few lines below, they state that natural polymer-based fibers enhance cell adhesion, and also consider it positive. So, should the fibers reduce or enhance the adhesion of cells? This issue needs clarification.

Round 2

Reviewer 1 Report

Comments and Suggestions for Authors

Dear Authors,

Thank you very much for your serious consideration of the observations made.

After reading your revised manuscript, it was noted that all observations and issues were addressed.

The flowchart of the literature search is highly appreciated, as it is found to be very helpful for an overview of the literature search, even if the authors did not perform a systematic review.